# A PAC-Bayesian Approach to Generalization Bounds for Graph Neural Networks

**Renjie Liao**[1,2]**, Raquel Urtasun**[1,2,3]**, Richard Zemel**[1,2,3]
University of Toronto[1], Vector Institute[3], Canadian Institute for Advanced Research[3]
{rjliao, urtasun, zemel}@cs.toronto.edu

## ABSTRACT

In this paper, we derive generalization bounds for two primary classes of graph neural networks (GNNs), namely graph convolutional networks (GCNs) and message passing GNNs (MPGNNs), via a PAC-Bayesian approach. Our result reveals that the maximum node degree and the spectral norm of the weights govern the generalization bounds of both models. We also show that our bound for GCNs is a natural generalization of the results developed in (Neyshabur et al., 2017) for fully-connected and convolutional neural networks. For MPGNNs, our PAC-Bayes bound improves over the Rademacher complexity based bound (Garg et al., 2020), showing a tighter dependency on the maximum node degree and the maximum hidden dimension. The key ingredients of our proofs are a perturbation analysis of GNNs and the generalization of PAC-Bayes analysis to non-homogeneous GNNs. We perform an empirical study on several synthetic and real-world graph datasets and verify that our PAC-Bayes bound is tighter than others.

## 1 INTRODUCTION

Graph neural networks (GNNs) (Gori et al., 2005; Scarselli et al., 2008; Bronstein et al., 2017; Battaglia et al., 2018) have become very popular recently due to their ability to learn powerful representations from graph-structured data, and have achieved state-of-the-art results in a variety of application domains such as social networks (Hamilton et al., 2017; Xu et al., 2018), quantum chemistry (Gilmer et al., 2017; Chen et al., 2019a), computer vision (Qi et al., 2017; Monti et al., 2017), reinforcement learning (Sanchez-Gonzalez et al., 2018; Wang et al., 2018), robotics (Casas et al., 2019; Liang et al., 2020), and physics (Henrion et al., 2017). Given a graph along with node/edge features, GNNs learn node/edge representations by propagating information on the graph via local computations shared across the nodes/edges. Based on the specific form of local computation employed, GNNs can be divided into two categories: graph convolution based GNNs (Bruna et al., 2013; Duvenaud et al., 2015; Kipf & Welling, 2016) and message passing based GNNs (Li et al., 2015; Dai et al., 2016; Gilmer et al., 2017). The former generalizes the convolution operator from regular graphs (*e.g.*, grids) to ones with arbitrary topology, whereas the latter mimics message passing algorithms and parameterizes the shared functions via neural networks.

Due to the tremendous empirical success of GNNs, there is increasing interest in understanding their theoretical properties. For example, some recent works study their expressiveness (Maron et al., 2018; Xu et al., 2018; Chen et al., 2019b), that is, what class of functions can be represented by GNNs. However, only few works investigate why GNNs generalize so well to unseen graphs. They are either restricted to a specific model variant (Verma & Zhang, 2019; Du et al., 2019; Garg et al., 2020) or have loose dependencies on graph statistics (Scarselli et al., 2018).

On the other hand, GNNs have close ties to standard feedforward neural networks, *e.g.*, multi-layer perceptrons (MLPs) and convolutional neural networks (CNNs). In particular, if each i.i.d. sample is viewed as a node, then the whole dataset becomes a graph without edges. Therefore, GNNs can be seen as generalizations of MLPs/CNNs since they model not only the regularities within a sample but also the dependencies among samples as defined in the graph. It is therefore natural to ask if we can generalize the recent advancements on generalization bounds for MLPs/CNNs (Harvey et al., 2017; Neyshabur et al., 2017; Bartlett et al., 2017; Dziugaite & Roy, 2017; Arora et al., 2018; 2019) to GNNs, and how would graph structures affect the generalization bounds?

In this paper, we answer the above questions by proving generalization bounds for the two primary classes of GNNs, *i.e.*, graph convolutional networks (GCNs) (Kipf & Welling, 2016) and message-passing GNNs (MPGNNs) (Dai et al., 2016; Jin et al., 2018).

Our generalization bound for GCNs shows an intimate relationship with the bounds for MLPs/CNNs with ReLU activations (Neyshabur et al., 2017; Bartlett et al., 2017). In particular, they share the same term, *i.e.*, the product of the spectral norms of the learned weights at each layer multiplied by a factor that is additive across layers. The bound for GCNs has an additional multiplicative factor $d^{(l-1)/2}$ where $d-1$ is the maximum node degree and $l$ is the network depth. Since MLPs/CNNs are special GNNs operating on graphs without edges (*i.e.*, $d-1=0$), the bound for GCNs coincides with the ones for MLPs/CNNs with ReLU activations (Neyshabur et al., 2017) on such degenerated graphs. Therefore, our result is a natural generalization of the existing results for MLPs/CNNs.

Our generalization bound for message passing GNNs reveals that the governing terms of the bound are similar to the ones of GCNs, *i.e.*, the geometric series of the learned weights and the multiplicative factor $d^{l-1}$. The geometric series appears due to the weight sharing across message passing steps, thus corresponding to the product term across layers in GCNs. The term $d^{l-1}$ encodes the key graph statistics. Our bound improves the dependency on the maximum node degree and the maximum hidden dimension compared to the recent Rademacher complexity based bound (Garg et al., 2020). Moreover, we compute the bound values on four real-world graph datasets (*e.g.*, social networks and protein structures) and verify that our bounds are tighter.

In terms of the proof techniques, our analysis follows the PAC-Bayes framework in the seminal work of (Neyshabur et al., 2017) for MLPs/CNNs with ReLU activations. However, we make two distinctive contributions which are customized for GNNs. First, a naive adaptation of the perturbation analysis in (Neyshabur et al., 2017) does not work for GNNs since ReLU is not 1-Lipschitz under the spectral norm, *i.e.*, $\|\text{ReLU}(X)\|_2 \leq \|X\|_2$ does not hold for some real matrix $X$. Instead, we construct the recursion on certain node representations of GNNs like the one with maximum $\ell_2$ norm, so that we can perform perturbation analysis with vector 2-norm. Second, in contrast to (Neyshabur et al., 2017) which only handles the homogeneous networks, *i.e.*, $f(ax) = af(x)$ when $a \geq 0$, we properly construct a quantity of the learned weights which 1) provides a way to satisfy the constraints of the previous perturbation analysis and 2) induces a finite covering on the range of the quantity so that the PAC-Bayes bound holds for all possible weights. This generalizes the analysis to non-homogeneous GNNs like typical MPGNNs.

The rest of the paper is organized as follows. In Section 2, we introduce background material necessary for our analysis. We then present our generalization bounds and the comparison to existing results in Section 3. We also provide an empirical study to support our theoretical arguments in Section 4. At last, we discuss the extensions, limitations and some open problems.

## 2 BACKGROUND

In this section, we first explain our analysis setup including notation and assumptions. We then describe the two representative GNN models in detail. Finally, we review the PAC-Bayes analysis.

### 2.1 ANALYSIS SETUP

In the following analysis, we consider the $K$-class graph classification problem which is common in the GNN literature, where given a graph sample $z$, we would like to classify it into one of the predefined $K$ classes. We will discuss extensions to other problems like graph regression in Section 5. Each graph sample $z$ is a triplet of an adjacency matrix $A$, node features $X \in \mathbb{R}^{n \times h_0}$ and output label $y \in \mathbb{R}^{1 \times K}$, *i.e.* $z = (A, X, y)$, where $n$ is the number of nodes and $h_0$ is the input feature dimension. We start our discussion by defining our notations. Let $\mathbb{N}_k^+$ be the first $k$ positive integers, *i.e.*, $\mathbb{N}_k^+ = \{1, 2, \ldots, k\}$, $|\cdot|_p$ the vector $p$-norm and $\|\cdot\|_p$ the operator norm induced by the vector $p$-norm. Further, $\|\cdot\|_F$ denotes the Frobenius norm of a matrix, $e$ the base of the natural logarithm function $\log$, $A[i, j]$ the $(i, j)$-th element of matrix $A$ and $A[i, :]$ the $i$-th row. We use parenthesis to avoid the ambiguity, *e.g.*, $(AB)[i, j]$ means the $(i, j)$-th element of the product matrix $AB$. We then introduce some terminologies from statistical learning theory and define the sample space as $\mathcal{Z}$, $z = (A, X, y) \in \mathcal{Z}$ where $X \in \mathcal{X}$ (node feature space) and $A \in \mathcal{G}$ (graph space), data distribution

$\mathcal{D}$, $z \overset{iid}{\sim} \mathcal{D}$, hypothesis (or model) $f_w$ where $f_w \in \mathcal{H}$ (hypothesis class), and training set $S$ with size $m$, $S = \{z_1, \ldots, z_m\}$. We make the following assumptions which also appear in the literature:

A1 Data, *i.e.*, triplets $(A, X, y)$, are i.i.d. samples drawn from some unknown distribution $\mathcal{D}$.

A2 The maximum hidden dimension across all layers is $h$.

A3 Node feature of any graph is contained in a $\ell_2$-ball with radius $B$. Specifically, we have $\forall i \in \mathbb{N}_n^+$, the $i$-th node feature $X[i,:] \in \mathcal{X}_{B,h_0} = \{x \in \mathbb{R}^{h_0} | \sum_{j=1}^{h_0} x_j^2 \le B^2\}$.

A4 We only consider simple graphs (*i.e.*, undirected, no loops[1], and no multi-edges) with maximum node degree as $d - 1$.

Note that it is straightforward to estimate $B$ and $d$ empirically on real-world graph data.

## 2.2 GRAPH NEURAL NETWORKS (GNNs)

In this part, we describe the details of the GNN models and the loss function we used for the graph classification problem. The essential idea of GNNs is to propagate information over the graph so that the learned representations capture the dependencies among nodes/edges. We now review two classes of GNNs, GCNs and MPGNNs, which have different mechanisms for propagating information. We choose them since they are the most popular variants and represent two common types of neural networks, *i.e.*, feedforward (GCNs) and recurrent (MPGNNs) neural networks. We discuss the extension of our analysis to other GNN variants in Section 5. For ease of notation, we define the model to be $f_w \in \mathcal{H} : \mathcal{X} \times \mathcal{G} \to \mathbb{R}^K$ where $w$ is the vectorization of all model parameters.

**GCNs:** Graph convolutional networks (GCNs) (Kipf & Welling, 2016) for the $K$-class graph classification problem can be defined as follows,

$$H_k = \sigma_k \left( \tilde{L} H_{k-1} W_k \right) \qquad (k\text{-th Graph Convolution Layer})$$

$$H_l = \frac{1}{n} \mathbf{1}_n H_{l-1} W_l \qquad (\text{Readout Layer}), \qquad (1)$$

where $k \in \mathbb{N}_{l-1}^+$, $H_k \in \mathbb{R}^{n \times h_k}$ are the node representations/states, $\mathbf{1}_n \in \mathbb{R}^{1 \times n}$ is a all-one vector, $l$ is the number of layers.[2] and $W_j$ is the weight matrix of the $j$-th layer. The initial node state is the observed node feature $H_0 = X$. For both GCNs and MPGNNs, we consider $l > 1$ since otherwise the model degenerates to a linear transformation which does not leverage the graph and is trivial to analyze. Due to assumption A2, $W_j$ is of size at most $h \times h$, *i.e.*, $h_k \le h$, $\forall k \in \mathbb{N}_{l-1}^+$. The graph Laplacian $\tilde{L}$ is defined as, $\tilde{A} = I + A$, $\tilde{L} = D^{-\frac{1}{2}} \tilde{A} D^{-\frac{1}{2}}$ where $D$ is the degree matrix of $\tilde{A}$. Note that the maximum eigenvalue of $\tilde{L}$ is 1 in this case. We absorb the bias into the weight by appending constant 1 to the node feature. Typically, GCNs use ReLU as the non-linearity, *i.e.*, $\sigma_i(x) = \max(0, x), \forall i = 1, \cdots, l-1$. We use the common mean-readout to obtain the graph representation where $H_{l-1} \in \mathbb{R}^{n \times h_{l-1}}$, $W_l \in \mathbb{R}^{h_{l-1} \times K}$, and $H_l \in \mathbb{R}^{1 \times K}$.

**MPGNNs:** There are multiple variants of message passing GNNs, *e.g.*, (Li et al., 2015; Dai et al., 2016; Gilmer et al., 2017), which share the same algorithmic framework but instantiate a few components differently, *e.g.*, the node state update function. We choose the same class of models as in (Garg et al., 2020) which are popular in the literature (Dai et al., 2016; Jin et al., 2018) in order to fairly compare bounds. This MPGNN model can be written in matrix forms as follows,

$$M_k = g(C_{\text{out}}^\top H_{k-1}) \qquad (k\text{-th step Message Computation})$$

$$\bar{M}_k = C_{\text{in}} M_k \qquad (k\text{-th step Message Aggregation})$$

$$H_k = \phi \left( X W_1 + \rho \left( \bar{M}_k \right) W_2 \right) \qquad (k\text{-th step Node State Update})$$

$$H_l = \frac{1}{n} \mathbf{1}_n H_{l-1} W_l \qquad (\text{Readout Layer}), \qquad (2)$$

---

[1] Here loop means an edge that connects a vertex to itself, a.k.a., self-loop.

[2] We count the readout function as a layer to be consistent with the existing analysis of MLPs/CNNs.

where $k \in \mathbb{N}_{l-1}^+$, $H_k \in \mathbb{R}^{n \times h_k}$ are node representations/states and $H_l \in \mathbb{R}^{1 \times K}$ is the output representation. Here we initialize $H_0 = \mathbf{0}$. W.l.o.g., we assume $\forall k \in \mathbb{N}_{l-1}^+$, $H_k \in \mathbb{R}^{n \times h}$ and $M_k \in \mathbb{R}^{n \times h}$ since $h$ is the maximum hidden dimension. $C_{\text{in}} \in \mathbb{R}^{n \times c}$ and $C_{\text{out}} \in \mathbb{R}^{n \times c}$ ($c$ is the number of edges) are the incidence matrices corresponding to incoming and outgoing nodes[3] respectively. Specifically, rows and columns of $C_{\text{in}}$ and $C_{\text{out}}$ correspond to nodes and edges respectively. $C_{\text{in}}[i, j] = 1$ indicates that the incoming node of the $j$-th edge is the $i$-th node. Similarly, $C_{\text{out}}[i, j] = 1$ indicates that the outgoing node of the $j$-th edge is the $i$-th node. $g, \phi, \rho$ are non-linear mappings, *e.g.*, ReLU and Tanh. Technically speaking, $g : \mathbb{R}^h \to \mathbb{R}^h$, $\phi : \mathbb{R}^h \to \mathbb{R}^h$, and $\rho : \mathbb{R}^h \to \mathbb{R}^h$ operate on vector-states of individual node/edge. However, since we share these functions across nodes/edges, we can naturally generalize them to matrix-states, *e.g.*, $\tilde{\phi} : \mathbb{R}^{n \times h} \to \mathbb{R}^{n \times h}$ where $\tilde{\phi}(X)[i, :] = \phi(X[i, :])$. By doing so, the same function could be applied to matrices with varying size of the first dimension. For simplicity, we use $g, \phi, \rho$ to denote such generalization to matrices. We denote the Lipschitz constants of $g, \phi, \rho$ under the vector 2-norm as $C_g, C_\phi, C_\rho$ respectively. We also assume $g(\mathbf{0}) = \mathbf{0}$, $\phi(\mathbf{0}) = \mathbf{0}$, and $\rho(\mathbf{0}) = \mathbf{0}$ and define the *percolation complexity* as $\mathcal{C} = C_g C_\phi C_\rho \|W_2\|_2$ following (Garg et al., 2020).

**Multiclass Margin Loss:** We use the multi-class $\gamma$-margin loss following (Bartlett et al., 2017; Neyshabur et al., 2017). The generalization error is defined as,

$$L_{\mathcal{D}, \gamma}(f_w) = \mathbb{P}_{z \sim \mathcal{D}} \left( f_w(X, A)[y] \le \gamma + \max_{j \ne y} f_w(X, A)[j] \right), \tag{3}$$

where $\gamma > 0$ and $f_w(X, A)$ is the $l$-th layer representations, *i.e.*, $H_l = f_w(X, A)$. Accordingly, we can define the empirical error as,

$$L_{S, \gamma}(f_w) = \frac{1}{m} \sum_{z_i \in S} \mathbf{1} \left( f_w(X, A)[y] \le \gamma + \max_{j \ne y} f_w(X, A)[j] \right). \tag{4}$$

## 2.3 BACKGROUND OF PAC-BAYES ANALYSIS

PAC-Bayes (McAllester, 1999; 2003; Langford & Shawe-Taylor, 2003) takes a Bayesian view of the probably approximately correct (PAC) learning theory (Valiant, 1984). In particular, it assumes that we have a prior distribution $P$ over the hypothesis class $\mathcal{H}$ and obtain a posterior distribution $Q$ over the same support through the learning process on the training set. Therefore, instead of having a deterministic model/hypothesis as in common learning formulations, we have a distribution of models. Under this Bayesian view, we define the generalization error and the empirical error as,

$$L_{S, \gamma}(Q) = \mathbb{E}_{w \sim Q}[L_{S, \gamma}(f_w)], \qquad L_{\mathcal{D}, \gamma}(Q) = \mathbb{E}_{w \sim Q}[L_{\mathcal{D}, \gamma}(f_w)].$$

Since many interesting models like neural networks are deterministic and the exact form of the posterior $Q$ induced by the learning process and the prior $P$ is typically unknown, it is unclear how one can perform PAC-Bayes analysis. Fortunately, we can exploit the following result from the PAC-Bayes theory.

**Theorem 2.1.** *(McAllester, 2003) (Two-sided) Let $P$ be a prior distribution over $\mathcal{H}$ and let $\delta \in (0, 1)$. Then, with probability $1 - \delta$ over the choice of an i.i.d. size-$m$ training set $S$ according to $\mathcal{D}$, for all distributions $Q$ over $\mathcal{H}$ and any $\gamma > 0$, we have*

$$L_{\mathcal{D}, \gamma}(Q) \le L_{S, \gamma}(Q) + \sqrt{\frac{D_{\text{KL}}(Q\|P) + \ln \frac{2m}{\delta}}{2(m-1)}}.$$

Here $D_{\text{KL}}$ is the KL-divergence. The nice thing about this result is that the inequality holds for all possible prior $P$ and posterior $Q$ distributions. Hence, we have the freedom to construct specific priors and posteriors so that we can work out the bound. Moreover, McAllester (2003); Neyshabur et al. (2017) provide a general recipe to construct the posterior such that for a large class of models, including deterministic ones, the PAC-Bayes bound can be computed. Taking a neural network as an example, we can choose a prior distribution with some known density, *e.g.*, a fixed Gaussian,

---

[3]For undirected graphs, we convert each edge into two directed edges.

over the initial weights. After the learning process, we can add random perturbations to the learned weights from another known distribution as long as the KL-divergence permits an analytical form. This converts the deterministic model into a distribution of models while still obtaining a tractable KL divergence. Leveraging Theorem 2.1 and the above recipe, Neyshabur et al. (2017) obtained the following result which holds for a large class of deterministic models.

**Lemma 2.2.** *(Neyshabur et al., 2017)[4] Let $f_w(x) : \mathcal{X} \to \mathbb{R}^K$ be any model with parameters $w$, and let $P$ be any distribution on the parameters that is independent of the training data. For any $w$, we construct a posterior $Q(w + u)$ by adding any random perturbation $u$ to $w$, s.t., $\mathbb{P}(\max_{x \in \mathcal{X}} |f_{w+u}(x) - f_w(x)|_\infty < \frac{\gamma}{4}) > \frac{1}{2}$. Then, for any $\gamma, \delta > 0$, with probability at least $1 - \delta$ over an i.i.d. size-$m$ training set $S$ according to $\mathcal{D}$, for any $w$, we have:*

$$L_{\mathcal{D},0}(f_w) \le L_{S,\gamma}(f_w) + \sqrt{\frac{2D_{\mathrm{KL}}(Q(w+u)\|P) + \log\frac{8m}{\delta}}{2(m-1)}}.$$

This lemma guarantees that, as long as the change of the output brought by the perturbations is small with a large probability, one can obtain the corresponding generalization bound.

## 3 GENERALIZATION BOUNDS

In this section, we present the main results: generalization bounds of GCNs and MPGNNs using a PAC-Bayesian approach. We then relate them to existing generalization bounds of GNNs and draw connections to the bounds of MLPs/CNNs. We summarize the key ideas of the proof in the main text and defer the details to the appendix.

### 3.1 PAC-BAYES BOUNDS OF GCNS

As discussed above, in order to apply Lemma 2.2, we must ensure that the change of the output brought by the weight perturbations is small with a large probability. In the following lemma, we bound this change using the product of the spectral norms of learned weights at each layer and a term depending on some statistics of the graph.

**Lemma 3.1.** *(GCN Perturbation Bound) For any $B > 0, l > 1$, let $f_w \in \mathcal{H} : \mathcal{X} \times \mathcal{G} \to \mathbb{R}^K$ be a $l$-layer GCN. Then for any $w$, and $x \in \mathcal{X}_{B,h_0}$, and any perturbation $u = vec(\{U_i\}_{i=1}^l)$ such that $\forall i \in \mathbb{N}_l^+, \|U_i\|_2 \le \frac{1}{l}\|W_i\|_2$, the change in the output of GCN is bounded as,*

$$|f_{w+u}(X, A) - f_w(X, A)|_2 \le eBd^{\frac{l-1}{2}}\left(\prod_{i=1}^l \|W_i\|_2\right)\sum_{k=1}^l \frac{\|U_k\|_2}{\|W_k\|_2}.$$

The key idea of the proof is to decompose the change of the network output into two terms which depend on two quantities of GNNs respectively: the maximum change of node representations $\max_i \left|H'_{l-1}[i,:] - H_{l-1}[i,:]\right|_2$ and the maximum node representation $\max_i |H_{l-1}[i,:]|_2$. Here the superscript prime denotes the perturbed model. These two terms can be bounded by an induction on the layer index. From this lemma, we can see that the most important graph statistic for the stability of GCNs is the maximum node degree, *i.e.*, $d - 1$. Armed with Lemma 3.1 and Lemma 2.2, we now present the PAC-Bayes generalization bound of GCNs as Theorem 3.2.

**Theorem 3.2.** *(GCN Generalization Bound) For any $B > 0, l > 1$, let $f_w \in \mathcal{H} : \mathcal{X} \times \mathcal{G} \to \mathbb{R}^K$ be a $l$ layer GCN. Then for any $\delta, \gamma > 0$, with probability at least $1 - \delta$ over the choice of an i.i.d. size-$m$ training set $S$ according to $\mathcal{D}$, for any $w$, we have,*

$$L_{\mathcal{D},0}(f_w) \le L_{S,\gamma}(f_w) + \mathcal{O}\left(\sqrt{\frac{B^2 d^{l-1}l^2 h \log(lh)\prod_{i=1}^l \|W_i\|_2^2 \sum_{i=1}^l(\|W_i\|_F^2/\|W_i\|_2^2) + \log\frac{ml}{\delta}}{\gamma^2 m}}\right).$$

---

[4]The constants slightly differ from the original paper since we use a two-sided version of Theorem 2.1.

Since it is easy to show GCNs are homogeneous, the proof of Theorem 3.2 follows the one for MLPs/CNNs with ReLU activations in (Neyshabur et al., 2017). In particular, we choose the prior distribution $P$ and the perturbation distribution to be zero-mean Gaussians with the same diagonal variance $\sigma$. The key steps of the proof are: 1) constructing a quantity of learned weights $\beta = (\prod_{i=1}^{l} \|W_i\|_2)^{1/l}$; 2) fixing any $\tilde{\beta}$, considering all $\beta$ that are in the range $|\beta - \tilde{\beta}| \leq \beta/l$ and choosing $\sigma$ which depends on $\tilde{\beta}$ so that one can apply Lemma 3.1 and 2.2 to obtain the PAC-Bayes bound; 3) taking a union bound of the result in the 2nd step by considering multiple choices of $\tilde{\beta}$ so that all possible values of $\beta$ (corresponding to all possible weight $w$) are covered. Although Lemma 2.2 and 3.1 have their own constraints on the random perturbation, above steps provide a way to set the variance $\sigma$ which satisfies these constraints and the independence w.r.t. learned weights. The latter is important since $\sigma$ is also the variance of the prior $P$ which should not depend on data.

## 3.2 PAC-BAYES BOUNDS OF MPGNNS

For MPGNNs, we again need to perform a perturbation analysis to make sure that the change of the network output brought by the perturbations on weights is small with a large probability. Following the same strategy adopted in proving Lemma 3.1, we prove the following Lemma.

**Lemma 3.3.** *(MPGNN Perturbation Bound) For any $B > 0, l > 1$, let $f_w \in \mathcal{H} : \mathcal{X} \times \mathcal{G} \to \mathbb{R}^K$ be a l-step MPGNN. Then for any $w$, and $x \in \mathcal{X}_{B,h_0}$, and any perturbation $u = vec(\{U_1, U_2, U_l\})$ such that $\eta = \max\left( \frac{\|U_1\|_2}{\|W_1\|_2}, \frac{\|U_2\|_2}{\|W_2\|_2}, \frac{\|U_l\|_2}{\|W_l\|_2} \right) \leq \frac{1}{l}$, the change in the output of MPGNN is bounded as,*

$$|f_{w+u}(X, A) - f_w(X, A)|_2 \leq eBl\eta\|W_1\|_2\|W_l\|_2 C_\phi \frac{(d\mathcal{C})^{l-1} - 1}{d\mathcal{C} - 1},$$

*where $\mathcal{C} = C_\phi C_\rho C_g \|W_2\|_2$.*

The proof again involves decomposing the change into two terms which depend on two quantities respectively: the maximum change of node representations $\max_i \left| H'_{l-1}[i, :] - H_{l-1}[i, :] \right|_2$ and the maximum node representation $\max_i |H_{l-1}[i, :]|_2$. Then we perform an induction on the layer index to obtain their bounds individually. Due to the weight sharing across steps, we have a form of geometric series $((d\mathcal{C})^{l-1} - 1)/(d\mathcal{C} - 1)$ rather than the product of spectral norms of each layer as in GCNs. Technically speaking, the above lemma only works with $d\mathcal{C} \neq 1$. We refer the reader to the appendix for the special case of $d\mathcal{C} = 1$. We now provide the generalization bound for MPGNNs.

**Theorem 3.4.** *(MPGNN Generalization Bound) For any $B > 0, l > 1$, let $f_w \in \mathcal{H} : \mathcal{X} \times \mathcal{G} \to \mathbb{R}^K$ be a l-step MPGNN. Then for any $\delta, \gamma > 0$, with probability at least $1 - \delta$ over the choice of an i.i.d. size-m training set $S$ according to $\mathcal{D}$, for any $w$, we have,*

$$L_{\mathcal{D},0}(f_w) \leq L_{S,\gamma}(f_w) + \mathcal{O}\left( \sqrt{\frac{B^2 \left(\max\left(\zeta^{-(l+1)}, (\lambda\xi)^{(l+1)/l}\right)\right)^2 l^2 h \log(lh)|w|_2^2 + \log \frac{m(l+1)}{\delta}}{\gamma^2 m}} \right),$$

*where $\zeta = \min\left( \|W_1\|_2, \|W_2\|_2, \|W_l\|_2 \right)$, $|w|_2^2 = \|W_1\|_F^2 + \|W_2\|_F^2 + \|W_l\|_F^2$, $\mathcal{C} = C_\phi C_\rho C_g \|W_2\|_2$, $\lambda = \|W_1\|_2\|W_l\|_2$, and $\xi = C_\phi \frac{(d\mathcal{C})^{l-1} - 1}{d\mathcal{C} - 1}$.*

The proof also contains three steps: 1) since MPGNNs are typically non-homogeneous, *e.g.*, when any of $\phi$, $\rho$, and $g$ is a bounded non-linearity like Sigmoid or Tanh, we design a special quantity of learned weights $\beta = \max(\zeta^{-1}, (\lambda\xi)^{1/l})$. 2) fixing any $\tilde{\beta}$, considering all $\beta$ that are in the range $|\beta - \tilde{\beta}| \leq \beta/(l + 1)$ and choosing $\sigma$ which depends on $\tilde{\beta}$ so that one can apply Lemma 3.3 and 2.2 to work out the PAC-Bayes bound; 3) taking a union bound of the previous result by considering multiple choices of $\tilde{\beta}$ so that all possible values of $\beta$ are covered. The case with $d\mathcal{C} = 1$ is again included in the appendix. The first step is non-trivial since we do not have the nice construction as in the homogeneous case, *i.e.*, normalizing the weights so that the spectral norms of weights across layers are the same while the network output is unchanged. Moreover, the quantity is vital to the whole proof framework since it determines whether one can 1) satisfy the constraints on the random perturbation (so that Lemma 2.2 and 3.3 are applicable) and 2) simultaneously induce a finite covering on its range (so that the bound holds for any $w$). Since it highly depends on the form of the perturbation bound and the network architecture, there seems to be no general recipe on how to construct such a quantity.

| Statistics | Max Node Degree $d-1$ | Max Hidden Dim $h$ | Spectral Norm of Learned Weights |
|---|---|---|---|
| VC-Dimension (Scarselli et al., 2018) | - | $\mathcal{O}\left(h^4\right)$ | - |
| Rademacher Complexity (Garg et al., 2020) | $\mathcal{O}\left(d^{l-1}\sqrt{\log(d^{2l-3})}\right)$ | $\mathcal{O}\left(h\sqrt{\log h}\right)$ | $\mathcal{O}\left(\lambda\mathcal{C}\xi\sqrt{\log\left(\|W_2\|_2\lambda\xi^2\right)}\right)$ |
| Ours | $\mathcal{O}\left(d^{l-1}\right)$ | $\mathcal{O}\left(\sqrt{h\log h}\right)$ | $\mathcal{O}\left(\lambda^{1+\frac{1}{l}}\xi^{1+\frac{1}{l}}\sqrt{\|W_1\|_F^2+\|W_2\|_F^2+\|W_l\|_F^2}\right)$ |

Table 1: Comparison of generalization bounds for GNNs. "-" means inapplicable. $l$ is the network depth. Here $\mathcal{C} = C_\phi C_\rho C_g\|W_2\|_2$, $\xi = C_\phi\frac{(d\mathcal{C})^{l-1}-1}{d\mathcal{C}-1}$, $\zeta = \min\left(\|W_1\|_2, \|W_2\|_2, \|W_l\|_2\right)$, and $\lambda = \|W_1\|_2\|W_l\|_2$. More details about the comparison can be found in Appendix A.5.

### 3.3 COMPARISON WITH OTHER BOUNDS

In this section, we compare our generalization bounds with the ones in the GNN literature and draw connections with existing MLPs/CNNs bounds.

#### 3.3.1 COMPARISON WITH EXISTING GNN GENERALIZATION BOUNDS

We compare against the VC-dimension based bound in (Scarselli et al., 2018) and the most recent Rademacher complexity based bound in (Garg et al., 2020). Our results are not directly comparable to (Du et al., 2019) since they consider a "infinite-wide" class of GNNs constructed based on the neural tangent kernel (Jacot et al., 2018), whereas we focus on commonly-used GNNs. Comparisons to (Verma & Zhang, 2019) are also difficult since: 1) they only show the bound for one graph convolutional layer, *i.e.*, it does not depend on the network depth $l$; and 2) their bound scales as $\mathcal{O}\left(\lambda_{\max}^{2T}/m\right)$, where $T$ is the number of SGD steps and $\lambda_{\max}$ is the maximum absolute eigenvalue of Laplacian $L = D - A$. Therefore, for certain graphs[5], the generalization gap is monotonically increasing with $T$, which cannot explain the generalization phenomenon. We compare different bounds by examining their dependency on three terms: the maximum node degree, the spectral norm of the learned weights, and the maximum hidden dimension. We summarize the overall comparison in Table 1 and leave the details such as how we convert bounds into our context to Appendix A.5.

**Max Node Degree** $(d-1)$**:**  The Rademacher complexity bound scales as $\mathcal{O}\left(d^{l-1}\sqrt{\log(d^{2l-3})}\right)$ whereas ours scales as $\mathcal{O}(d^{l-1})$[6]. Many real-world graphs such as social networks tend to have large hubs (Barabási et al., 2016), which lead to very large node degrees. Thus, our bound would be significantly better in these scenarios. It is noteworthy that if one further introduces some assumption, *e.g.*, $\phi$ is a squashing function like tanh as shown in (Garg et al., 2020), then one can improve the above exponential dependency on the network depth $l$ for both Rademacher complexity and PAC-Bayes bounds.

**Max Hidden Dimension** $h$**:**  Our bound scales as $\mathcal{O}(\sqrt{h\log h})$ which is tighter than the Rademacher complexity bound $\mathcal{O}\left(h\sqrt{\log h}\right)$ and the VC-dimension bound $\mathcal{O}(h^4)$.

**Spectral Norm of Learned Weights:**  As shown in Table 1, we cannot compare the dependencies on the spectral norm of learned weights without knowing the actual values of the learned weights. Therefore, we perform an empirical study in Section 4.

#### 3.3.2 CONNECTIONS WITH EXISTING BOUNDS OF MLPs/CNNs

As described above, MLPs/CNNs can be viewed as special cases of GNNs. In particular, we have two ways to show the inclusion relationship. First, we can treat each i.i.d. sample as a node and the whole dataset as a graph without edges. Then conventional tasks (*e.g.*, classification) become node-level tasks (*e.g.*, node classification) on this graph. Second, we can treat each i.i.d. sample as a single-node graph. Then conventional tasks (*e.g.*, classification) becomes graph-level tasks (*e.g.*, graph classification). Since we focus on the graph classification, we adopt the second view. In

---

[5]Since $\lambda_{\max} = \max_{v\neq\mathbf{0}}(v^\top(D-A)v)/(v^\top v)$, we have $\lambda_{\max} \geq (D-A)[i,i]$ by choosing $v = e_i$, *i.e.*, the $i$-th standard basis. We can pick any node $i$ which has more than 1 neighbor to make $\lambda_{\max} > 1$.

[6]Our bound actually scales as $\mathcal{O}\left(d^{(l+1)(l-2)/l}\right)$ which is upper bounded by $\mathcal{O}\left(d^{l-1}\right)$.

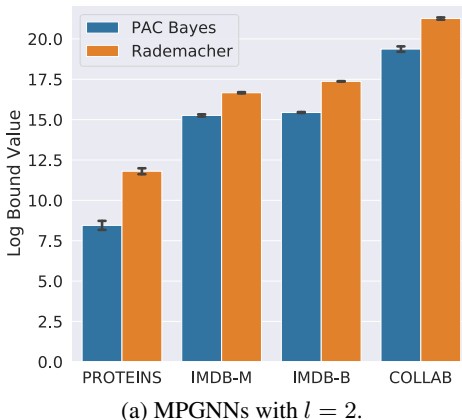 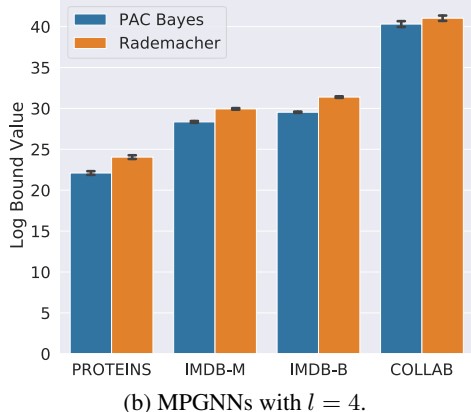

(a) MPGNNs with $l = 2$.

(b) MPGNNs with $l = 4$.

Figure 1: Bound evaluations on real-world datasets. The maximum node degrees (*i.e.*, $d-1$) of four datasets from left to right are: $25$ (PROTEINS), $88$ (IMDB-M), $135$ (IMDB-B), and $491$ (COLLAB).

particular, MLPs/CNNs with ReLU activations are equivalent to GCNs with the graph Laplacian $\tilde{L} = I$ (hence $d = 1$). We leave the details of this conversion to Appendix A.6. We restate the PAC-Bayes bound for MLPs/CNNs with ReLU activations in (Neyshabur et al., 2017) as follows,

$$L_{\mathcal{D},0}(f_w) \leq L_{S,\gamma}(f_w) + \mathcal{O}\left(\sqrt{\left(B^2 l^2 h \log(lh) \prod_{i=1}^{l} \|W_i\|_2^2 \sum_{i=1}^{l} (\|W_i\|_F^2/\|W_i\|_2^2) + \log\frac{ml}{\delta}\right)/\gamma^2 m}\right).$$

Comparing it with our bound for GCNs in Theorem 3.2, it is clear that we only add a factor $d^{l-1}$ to the first term inside the square root which is due to the underlying graph structure of the data. If we apply GCNs to single-node graphs, the two bounds coincide since $d = 1$. Therefore, our Theorem 3.2 directly generalizes the result in (Neyshabur et al., 2017) to GCNs, which is a strictly larger class of models than MLPs/CNNs with ReLU activations.

## 4 EXPERIMENTS

In this section, we perform an empirical comparison between our bound and the Rademacher complexity bound for MPGNNs. We experiment on 6 synthetic datasets of random graphs (corresponding to 6 random graph models), 3 social network datasets (COLLAB, IMDB-BINARY, IMDB-MULTI), and a bioinformatics dataset PROTEINS from (Yanardag & Vishwanathan, 2015). In particular, we create synthetic datasets by generating random graphs from the Erdős–Rényi model and the stochastic block model with different settings (*i.e.*, number of blocks and edge probabilities). All datesets focus on graph classifications. We repeat all experiments 3 times with different random initializations and report the means and the standard deviations. Constants are considered in the bound computation. More details of the experimental setup, dataset statistics, and the bound computation are provided in Appendix A.7.

As shown in Fig. 1 and Fig. 2, our bound is mostly tighter than the Rademacher complexity bound with varying message passing steps $l$ on both synthetic and real-world datasets. Generally, the larger the maximum node degree is, the more our bound improves[7] over the Rademacher complexity bound (*c.f.*, PROTEINS vs. COLLAB). This could be attributed to the better dependency on $d$ of our bound. For graphs with large node degrees (*e.g.*, social networks like Twitter have influential users with lots of followers), the gap could be more significant. Moreover, with the number of steps/layers increasing, our bound also improves more in most cases. It may not be clear to read from the figures since the y-axis is in the log domain and its range differ from figure to figure. We also provide the numerical values of the bound evaluations in the appendix for an exact comparison. The number of steps is chosen to be no larger than 10 as GNNs are generally shown to perform well with just a few steps/layers (Kipf & Welling, 2016; Jin et al., 2018). We found $d\mathcal{C} > 1$ and the geometric series $((d\mathcal{C})^{l-1}-1)/(d\mathcal{C}-1) \gg 1$ on all datasets which imply learned GNNs are not contraction mappings (*i.e.*, $d\mathcal{C} < 1$). This also explains why both bounds becomes larger with more steps. At last, we can see that bound values are much larger than 1 which indicates both bounds are still vacuous, similarly to the cases for regular neural networks in (Bartlett et al., 2017; Neyshabur et al., 2017).

---

[7]Note that it may not be obvious from the figure as the y axis is in log domain. Please refer to the appendix where the actual bound values are listed in the table.

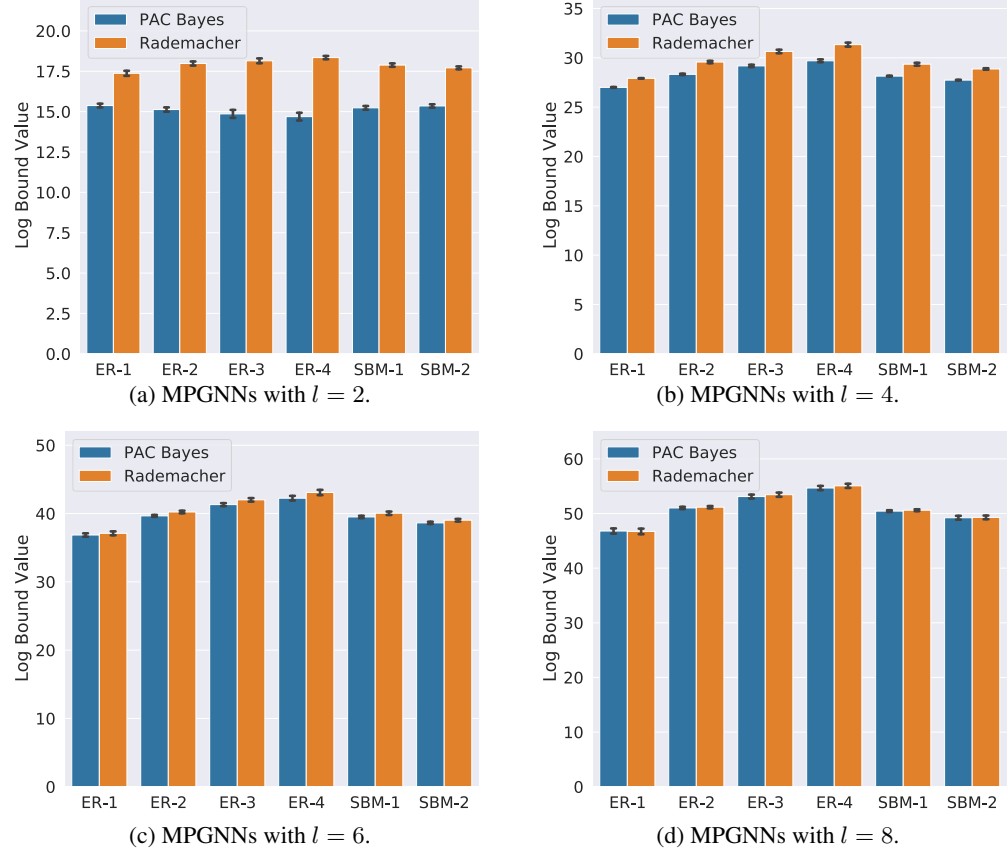

Figure 2: Bound evaluations on synthetic datasets. The maximum node degrees (*i.e.*, $d - 1$) of datasets from left to right are: $25$ (ER-1), $48$ (ER-2), $69$ (ER-3), $87$ (ER-4), $25$ (SBM-1), and $36$ (SBM-2). 'ER-X' and 'SBM-X' denote the Erdős–Rényi model and the stochastic block model with the 'X'-th setting respectively. Please refer to the appendix for more details.

## 5 DISCUSSION

In this paper, we present generalization bounds for two primary classes of GNNs, *i.e.*, GCNs and MPGNNs. We show that the maximum node degree and the spectral norms of learned weights govern the bound for both models. Our results for GCNs generalize the bounds for MLPs/CNNs in (Neyshabur et al., 2017), while our results for MPGNNs improve over the state-of-the-art Rademacher complexity bound in (Garg et al., 2020). Our PAC-Bayes analysis can be generalized to other graph problems such as node classification and link prediction since our perturbation analysis bounds the maximum change of any node representation. Other loss functions (*e.g.*, ones for regression) could also work in our analysis as long as they are bounded.

However, we are far from being able to explain the practical behaviors of GNNs. Our bound values are still vacuous as shown in the experiments. Our perturbation analysis is in the worst-case sense which may be loose for most cases. We introduce Gaussian posterior in the PAC-Bayes framework to obtain an analytical form of the KL divergence. Nevertheless, the actual posterior induced by the prior and the learning process may likely to be non-Gaussian. We also do not explicitly consider the optimization algorithm in the analysis which clearly has an impact on the learned weights.

This work leads to a few interesting open problems for future work: (1) Is the maximum node degree the only graph statistic that has an impact on the generalization ability of GNNs? Investigating other graph statistics may provide more insights on the behavior of GNNs and inspire the development of novel models and algorithms. (2) Would the analysis still work for other interesting GNN architectures, such as those with attention (Veličković et al., 2017) and learnable spectral filters (Liao et al., 2019)? (3) Can recent advancements for MLPs/CNNs, *e.g.*, the compression technique in (Arora et al., 2018) and data-dependent prior of (Parrado-Hernández et al., 2012), help further improve the bounds for GNNs? (4) What is the impact of the optimization algorithms like SGD on the generalization ability of GNNs? Would graph structures play a role in the analysis of optimization?

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
