# OpenReview forum: "A PAC-Bayesian Approach to Generalization Bounds for Graph Neural Networks"
_ICLR.cc/2021/Conference — ICLR 2021 Poster_

### Official Review · AnonReviewer3 · 2020-10-25
**Generalization Bounds for Graph Neural Networks**

**Rating:** 6
**Confidence:** 3

**Review:**

This paper, by PAC-Bayesian approach, proves the generalization bounds for the two primary classes of graph neural networks—graph convolutional networks and message passing GNNs. By experiments on four datasets, it shows that the generalization bound in this paper is tighter than the existing Rademacher complexity bound.

This is an interesting question about generalization bound, especially, such a discussion by  PAC-bayesian approach for graph neural networks is lacking. However, there are a few issues/comments with the work:

1.The proof techniques in this paper are mainly from Neyshabur et al., 2017. The theoretical contribution and novelty are limited;

2.For the generalization bounds, they are exponential dependence on depth. So for $d>1$, it means that the deeper it is, the worse it is for the graphical neural network. Is that so?

3.It will be better to use more empirical experiments  to verify the relationship between the generalization bounds, node degrees, and depth;

4.For Assumption 4 in this paper, it assumes that “no loop”, for the practical application of graph neural networks, loop generally exists. This will limit the potential application of this theory. in this paper;

5.According to the theory established in this paper and the # graphs and max # nodes of the datasets provided in APPENDIX, PROTEINS and IMDB-BINARY should have more similar log bound value than IMDB-BINARY and MDB-MULTI. In Figure 1, it seems to be the opposite, why? Is it because the feature dimension also plays an important role in the log bound value?

6.The format of mathematical equations is inconsistent, for example, equation (1) without punctuation,  but equation (3) with punctuation.

Overall, I think this is an interesting piece of work that might interest researchers to explore the  questions around graph neural networks. However, I think the results need to be analyzed more carefully, especially on the depth. Moreover, the novelty of the technology is relatively limited.

---

> ### Author Response · Authors · 2020-11-19
> **Response to Reviewer 3**
>
> We thank the reviewer for the valuable and constructive feedback. We respond to the individual questions as below.
>
>
> Q1: The proof techniques in this paper are mainly from Neyshabur et al., 2017. The theoretical contribution and novelty are limited.
>
>
> > A1: Please refer to A1 in the common response.
>
>
> Q2: For the generalization bounds, they are exponential dependence on depth. So for d>1, it means that the deeper it is, the worse it is for the graphical neural network. Is that so?
>
>
> > A2: The dependency on the depth is indeed exponential which is also the case for the recent Rademacher complexity results [1]. This indicates the deeper the network is, the looser the bound is for d > 1. But it does not necessarily mean the worse the GNN performs since the uniform convergence bounds hold for all models in the class thus including the worst case (so does our perturbation analysis). As we discussed in the paper, the current bound is vacuous and can not fully explain the practical behaviors of GNNs.
> >
> >
> > [1] Garg, V.K., Jegelka, S. and Jaakkola, T., 2020. Generalization and representational limits of graph neural networks. In ICML.
>
>
> Q3: It will be better to use more empirical experiments to verify the relationship between the generalization bounds, node degrees, and depth.
>
>
> > A3: Thanks for the suggestion. We added new experiments with varying random graph families, node degrees, and depth. Please refer to A2 in the common response and the updated paper.
>
>
> Q4. For Assumption 4 in this paper, it assumes that “no loop”, for the practical application of graph neural networks, loop generally exists. This will limit the potential application of this theory in this paper.
>
>
> > A4: Sorry that we should have clarified the terminology. We use the graph theory definition, i.e., a loop means an edge that connects a vertex to itself (sometimes called self-loop). This is easily confused with cycles in graph theory which many people in practice also refer to as loops. We clarify them in the new version. Therefore, our results hold for graphs with “loops”. Moreover, self-loops can be easily handled in the perturbation analysis and they only affect the max node degree “d-1” in our bound. But multi-edges may require more subtle treatment.
>
>
> Q5: According to the theory established in this paper and the # graphs and max # nodes of the datasets provided in APPENDIX, PROTEINS and IMDB-BINARY should have more similar log bound value than IMDB-BINARY and IMDB-MULTI. In Figure 1, it seems to be the opposite, why? Is it because the feature dimension also plays an important role in the log bound value?
>
>
> > A5: We want to clarify that 1) our bounds do not depend on max # nodes; 2) our bounds depend on the feature dimension h in $O( \sqrt{ h \log h } )$; 3) the most significant term in our bounds is the max node degree. If we rank datasets ascendingly by the max node degree, they are PROTEINS, IMDB-MULTI, IMDB-BINARY, and COLLAB. This order mostly matches what we observed in terms of the ascending ranking of bound values in Figure 1.
>
>
> Q6: The format of mathematical equations is inconsistent, for example, equation (1) without punctuation, but equation (3) with punctuation.
>
>
> > A6: Thanks for pointing this out. We have corrected them in the new version.

---

### Official Review · AnonReviewer1 · 2020-10-26
**A good theory paper.**

**Rating:** 7
**Confidence:** 4

**Review:**

##########################################################################
Summary:
This paper provides results of generalization bounds for two types of GNNs: GCN and MPGNN. The presented analysis follows the framework of Neyshabur 2017 to construct posterior by adding random perturbations so that the PAC-Bayesian technique can be applied. The main contributions are the perturbation analysis for GCN and MPGNN, which results in a bound depending on the graph statistics. The paper compares the derived bounds with existing results, and examines the bounds numerically through experiments.

##########################################################################
I find the paper well-written and overall technically sound. I vote for accept.
##########################################################################

Strength:

This paper studies an important problem, as GNNs are popular learning models.

While the overall idea follows directly from Neyshabur 2017, the presented perturbation analysis is not trivial, making the current work a good advance.

This paper is well-organized, with nice discussions on the state-of-the-art as well as on the overall technical review. It is appreciated that the paper is self-contained, and the preliminary results are provided in the appendix (though they can be found in earlier papers).

The paper provides a detailed comparison between the derived bounds with the existing bounds. The results of this paper are natural in the sense they generalize that the results in Neyshabur 2017 for MLPs.


Some concerns:

For the proofs of Theorem 3.2 and 3.4, I would suggest decomposing it into several lemmas, by first identifying the covering (and the weight-independent variance) and then plugging the values into Lemma 3.3 (and Lemma 3.1) and further into Lemma 2.2

To avoid ambiguity, using D_KL(u+w||P) is better than D_KL(Q||P) in this paper.

Given the heavy notations, having a notation table in the appendix could be very helpful.

What are the key difficulties in applying the proposed framework to other GNN architectures?

---

> ### Author Response · Authors · 2020-11-19
> **Response to Reviewer 1**
>
> We thank the reviewer for the valuable and constructive feedback. We respond to the individual questions as below.
>
>
> Q1: For the proofs of Theorem 3.2 and 3.4, I would suggest decomposing it into several lemmas, by first identifying the covering (and the weight-independent variance) and then plugging the values into Lemma 3.3 (and Lemma 3.1) and further into Lemma 2.2. To avoid ambiguity, using D_KL(u+w||P) is better than D_KL(Q||P) in this paper. Given the heavy notations, having a notation table in the appendix could be very helpful.
>
>
> > A1: Thanks for your great suggestions! We improved the presentation and added the notation table in the new version. We will reorganize the proof to make it more clear.
>
>
> Q2: What are the key difficulties in applying the proposed framework to other GNN architectures?
>
>
> > A2: Given a new GNN architecture, there are mainly two difficulties in applying the framework.
> >
> >
> > First, developing the perturbation analysis. Some nonlinear mechanisms within GNNs like attention or spectral filters may raise some challenges in obtaining a tight perturbation bound. One may need to resort to more advanced inequalities or techniques.
> >
> >
> > Second, once the perturbation bound was established, the challenge is to construct a proper quantity of weights w so that 1) it makes all conditions on the random perturbations hold, e.g., the ones in Lemma 2.2, 3.1, and 3.3; 2) it induces a finite covering of its range. Since this part highly depends on the form of the perturbation bound and the network architecture, there seems to be no general recipe on how to construct such a quantity. Please refer to A1 in the common response for more discussion on this point.

---

### Official Review · AnonReviewer2 · 2020-10-29
**Official Blind Review #2**

**Rating:** 7
**Confidence:** 2

**Review:**

In this paper, the authors propose generalization bounds for GNNs, both convolutional and standard message passing variants. The result is a generalization of those for CNN/MLP architectures with relu activation functions. The analysis method closely follows those established in the former settings as well. The specific setting they consider is where each sample in the dataset is a graph. The proof relies on ensuring small perturbations in the GNN weights don't cause large deviations in output distributions. The resulting PAC-Bayes bound is shown to be tighter than corresponding Rademacher Complexity bounds.

The paper is well written and the results look reasonable (though I didn't check the proofs). A couple of minor comments/questions:

1. Does assumption A4 cause things to be a lot looser than needed? Consider a star graph, then d ~ number of nodes in the graph. But this is just for one node.

2. How would things change if your data is actually not iid ~ D, but the entire dataset is the graph (so you just have one graph), and you need to classify the nodes? I'm guessing there's parts in the proof where the iid assumption is necessary.

---

> ### Author Response · Authors · 2020-11-19
> **Response to Reviewer 2**
>
> We thank the reviewer for the valuable and constructive feedback. We respond to the individual questions as below.
>
>
>
>
> Q1: Does assumption A4 cause things to be a lot looser than needed? Consider a star graph, then d ~ number of nodes in the graph. But this is just for one node.
>
>
> > A1: The assumption A4 is needed since our perturbation analysis is in the worst-case sense, i.e., the change of any node representation is upper bounded under certain perturbations on the weights of GNN. If the distribution of node degrees is assumed to be known, then we can perform a probabilistic perturbation analysis which could overcome this issue. Alternatively, if the readout function only relies on a subgraph, e.g., applying some sort of hard attention mechanism, then this issue could also be alleviated.
>
>
>
> Q2: How would things change if your data is actually not iid ~ D, but the entire dataset is the graph (so you just have one graph), and you need to classify the nodes? I'm guessing there's parts in the proof where the iid assumption is necessary.
>
>
> > A2: Thanks for raising this good point! We did think about the semi-supervised node classification scenario as you mentioned. However, we did not investigate it further due to the space constraint. Intuitively, GNNs unroll a computation tree (depth corresponds to the message passing step / graph convolution layer) from a node perspective. Our perturbation analysis already bounds the change of the root representation of any such computation tree. So we could reuse this part. But in order to establish the full PAC-Bayes analysis, you still need an iid assumption of some form, e.g., computation trees of all nodes within this graph are iid. This is due to the fact that general PAC-Bayes results (Theorem 2.1 and Lemma 2.2) require such an assumption.

---

### Official Review · AnonReviewer4 · 2020-10-29
**Weak theoretical contributions**

**Rating:** 5
**Confidence:** 4

**Review:**

# Summary

The paper presents PAC-Bayesian generalization bounds for two classes of graph neural networks: graph convolutional neural networks and message passing graph neural networks. The paper essentially adapts Neyshabur et al. (2017) PAC-Bayesian margin bounds for neural networks to graph neural networks and expectedly the bounds contain terms that depend on the degree of the underlying graph. The main technical contribution of the paper is a perturbation bound for GNNs from which the main results follow.

# Strengths

1. The paper presents the first PAC-Bayesian generalization bound for GNNs and the authors show that their bounds are tighter than the Rademacher based generalization bounds developed by Garg et al (2020) ignoring constants.

# Weakness

1. The bounds do not necessarily give important insights into generalization performance of GNNs. Bounds are still vacuous: bounds become exponentially large with number of layers and degree.
2. For a theory paper that purely focuses on obtaining generalization bounds for GNNs, the technical contributions are weak. The only technical contribution is the derivation of perturbation bounds for GNNs which are pretty easy to obtain. If this is not the case, then this needs to be highlighted.
3. It is not sufficient to compare bounds on real world data at some fixed sample size for empirical comparison with Rademacher based bounds, especially given that constants are dropped. Synthetic experiments with varying samples, graph families (e.g. Erdos Renyi graphs) and different degree distribution are needed to show how the bounds compare against Rademacher based bounds.

# Justification for rating

The generalization bounds themselves provide very limited insights into generalization performance of GNNs especially given recent results that show that uniform convergence bounds may not be able to explain generalization of deep neural networks. While this is an issue with existing theoretical results for deep neural networks, the paper does not significantly improve the state-of-the-art in theoretical understanding of GNNs in terms of new tools and proof techniques. For a theory paper that purely focuses on generalization bounds, this is a significant shortcoming.

# Other comments

1. Misleading use of the word "statistics" throughout the paper. Statistics are quantities that can be computed only from the data. The paper repeatedly refers to functionals of parameters as "statistics".
2. What do you mean by: "actual posterior distribution induced by learning process may be very different from Gaussians" ? PAC-Bayesian analysis is done for Gibbs classifiers and Neyshabur et al. provide a way to convert these convergence bounds for deterministic classifiers. The learning process does not induce a posterior distribution over weights (assuming deterministic initialization and removing randomness like dropout).
3. D is overloaded to denote both data distribution and diagonal degree matrix.

---

> ### Author Response · Authors · 2020-11-19
> **Response to Reviewer 4**
>
> We thank the reviewer for the valuable and constructive feedback. We respond to the individual questions as below.
>
>
> Q1: The bounds do not necessarily give important insights into generalization performance of GNNs especially given recent results that show that uniform convergence bounds may not be able to explain generalization of deep neural networks. Bounds are still vacuous: bounds become exponentially large with the number of layers and degree.
>
>
> > A1: Please refer to A1 in the common response.
>
>
> Q2: The paper does not significantly improve the state-of-the-art in theoretical understanding of GNNs in terms of new tools and proof techniques. For a theory paper that purely focuses on obtaining generalization bounds for GNNs, the technical contributions are weak. The only technical contribution is the derivation of perturbation bounds for GNNs which are pretty easy to obtain. If this is not the case, then this needs to be highlighted.
>
>
> > A2: We appreciate your high standard in evaluating technical contributions. Please refer to A1 in the common response.
>
>
> Q3: It is not sufficient to compare bounds on real-world data at some fixed sample size for empirical comparison with Rademacher based bounds, especially given that constants are dropped. Synthetic experiments with varying samples, graph families (e.g. Erdos Renyi graphs) and different degree distributions are needed to show how the bounds compare against Rademacher based bounds.
>
>
> > A3: Thanks for the suggestion. We added experiments which consider the constants in bound evaluations and include 6 synthetic random graph families in the new version. Please refer to A2 in the common response, the updated paper, and the updated appendix.
>
>
> Q4: Misleading use of the word "statistics" throughout the paper. Statistics are quantities that can be computed only from the data. The paper repeatedly refers to functionals of parameters as "statistics".
>
>
> > A4: We appreciate your rigorousness w.r.t. terminologies. For clarification, we use the term statistic to refer to some function of both data and weights, e.g., the maximum node representations. Similar usage has become common in machine learning, e.g., “batch statistics” such as the mean of hidden representations in batch normalization [1] are functions of both data and weights. But we agree that it is confusing as people typically map the terminology weights in machine learning to parameters in statistical inference. We modified our wording in the new version.
> >
> > [1] Ioffe, S. and Szegedy, C., 2015. Batch normalization: Accelerating deep network training by reducing internal covariate shift. In ICML.
>
>
> Q5: What do you mean by: "actual posterior distribution induced by the learning process may be very different from Gaussians" ? PAC-Bayesian analysis is done for Gibbs classifiers and Neyshabur et al. provide a way to convert these convergence bounds for deterministic classifiers. The learning process does not induce a posterior distribution over weights (assuming deterministic initialization and removing randomness like dropout).
>
>
> > A5: Sorry for the confusion as we did not have space to expand the argument in detail in the submission. The results of Neyshabur et al. indeed provide a way to convert these convergence bounds for deterministic classifiers (their result is actually a generalization of an earlier result by David McAllester). They achieve it by adding Gaussian perturbations on the learned weights of the deterministic model.
> >
> >
> > Here we refer to another stochastic view of a deterministic model where people in practice often randomly initialize weights following a prior distribution, e.g., Gaussian or uniform, before learning. If we collect the learned weights returned by the learning process under different initial weights drawn from the prior distribution, then we can view them as drawn from a posterior distribution induced by the prior and the learning process. This posterior may be quite non-Gaussian, e.g., they may be a multimodal distribution since the learned weights may land in different local minima with different initialized weights. Therefore, the Gaussian perturbation may hardly capture this phenomenon.
> >
> >
> > This view follows more closely to how people actually train deterministic models in practice. Therefore, it may be worthwhile to investigate whether one can develop a PAC-Bayes result under this stochastic view. Of course, there are many challenges, e.g., the density of the induced distribution is unknown so that one can not obtain the KL divergence.
>
>
> Q6: D is overloaded to denote both data distribution and diagonal degree matrix.
>
>
> > A6: Thanks for pointing it out. We have corrected it in the new version.

---

### Author Response · Authors · 2020-11-19
**Common Response [1/2]**

We thank all the reviewers for their valuable feedback. We respond to some shared concerns or questions here.

Q1: Weak theoretical contributions due to limited technical contributions.

A1: We argue that proposing new tools/techniques is not the only way to make important theoretical contributions. Although our proof techniques follow Neyshabur et al. which are originally designed for deep ReLU networks, we did contribute better theoretical understandings of GNNs. In the following, we will clarify our theoretical and technical contributions.

**Theoretical Contributions**

Our work provides a few important insights into GNNs.


1. Although the max-node-degree is expected to play a role in the generalization bounds of GNNs, its specific function form matters. We show that our PAC-Bayesian approach gives a tighter dependence on it for message passing GNNs (MPGNNs) than Rademacher one: $O(d^{l-1})$ vs. $O(d^{l-1} \sqrt{ log(d^{2l-3}) })$. This factor becomes more significant in graphs with large degrees like hubs in social networks. This does tell us that different generalization bound approaches make a difference for GNNs. In addition, our bounds are more succinct than previous ones.


2. We explicitly reveal the relationship between the generalization bounds of GNNs and MLPs/CNNs, i.e., the PAC-Bayes bounds for MLPs/CNNs are special cases of ones for GNNs, which have not been discovered or stated previously in the literature. This relationship also matches closely to the observation that MLPs/CNNs can be regarded as special GNNs.


3. As discussed in Sec. 5, we agree that our bounds are still vacuous. But we believe they represent progress towards this goal for the GNN community, given the above insights and the additional facts that a) classes of GNNs we considered, i.e. GCNs and MPGNNs, are very popular in practice and are richer than ones considered in the existing literature; b) the area of generalization bounds for GNNs is largely underexplored (only ~4 papers exist to the best of our knowledge).


**Technical Contributions**


Our technical contributions include not only the perturbation analysis of two popular classes of GNNs but also a way of generalizing PAC-Bayes analysis to non-homogeneous GNNs.


1. The perturbation analysis itself is non-trivial since details of GNN architectures matter. In contrast to the one in Neyshabur et al. which only analyzes the perturbation per sample (i.e., per node in our context), we need to first identify and then solve two recursions of the maximum node representation and the maximum difference of node representations. Moreover, MPGNNs have more complicated architectures than GCNs and regular MLPs/CNNs which require more subtle and careful handling of recursions and inequalities in order to arrive at succinct forms. Finally, the perturbation results and analysis techniques could be useful for analyzing the robustness of GNNs on their own.


2. We generalize PAC-Bayes analysis to non-homogeneous networks that haven't been studied in Neyshabur et al.
\
\
 In particular, once the perturbation analysis was established, in order to arrive at the PAC-Bayes bound, one needs to 1) satisfy the constraints on the random perturbation in Lemma 2.2, 3.1, and 3.3; 2) consider all possible learned weights w and then take a union bound (since Theorem 3.2 and 3.4 hold for any w). To achieve this, a properly constructed quantity of w is vital so that it induces the constraints-satisfied variance of the perturbation and has a finite covering of its range (so that union bound is tractable and the variance is data-independent). For homogeneous networks (e.g., those with ReLU but not with Tanh or other bounded nonlinearities), one could follow the construction in Neyshabur et al.
\
\
 However, since MPGNNs are typically non-homogeneous, one can not normalize the weights to make the spectral norm of weights across layers the same while keeping the output of the network unchanged. Therefore, it is non-trivial to construct such a quantity to satisfy all inequalities and induce a finite covering since it highly depends on the form of the perturbation bound and the network architecture.
\
\
 Inspired by the construction of Neyshabur et al., we conjecture that the quantity should have a form so that if we assume the spectral norm of weights across layers are the same, its exponent does not depend on the network depth. Starting from this guess, we work in reverse by assuming the perturbation bound holds and check how to satisfy the conditions. After quite some guess-and-verify, we successfully construct one with a finite covering. Once we had this construction which is the most challenging step, the remaining PAC-Bayes analysis is somewhat straightforward.
\
\
 Since the above details were mostly embedded in the proof of the appendix, we realize that these technical contributions were not clear in the current paper. We briefly highlight them in the main paper in the new version.

---

### Author Response · Authors · 2020-11-19
**Common Response [2/2]**

Q2: More empirical comparisons of different bounds.

A2: Based on the suggestions from reviewers, we add the following experiments and report the mean and std of three runs with different random seeds under all cases. We also updated the paper to include more details and figures.

(1) **Adding constants**

We first add the corresponding constants to both bounds (see the updated appendix for details). The log bound values are shown in the table below.

|  **l=2**   |`     PROTEINS   `|`     IMDB-MULTI   `|`   IMDB-BINARY   `    |  `   COLLAB   `    |
|------------|:----------------:|:------------------:|:---------------------:|:------------------:|
|PAC-Bayes   | **8.19** $\pm$ 0.06 | **9.07** $\pm$ 0.03 | **9.57**$\pm$0.07 | **10.94** $\pm$ 0.06 |
|Rademacher  | 10.18 $\pm$ 0.03 | 11.82 $\pm$ 0.02 | 12.48 $\pm$ 0.04 | 14.15 $\pm$ 0.04 |
| **l=4** |  |  |  |  |
| PAC-Bayes  | **14.07** $\pm$ 0.06 | **16.92** $\pm$ 0.04 | **18.17** $\pm$ 0.06 | **24.53** $\pm$ 0.03 |
| Rademacher | 14.65 $\pm$ 0.05 | 18.15 $\pm$ 0.05 | 19.52 $\pm$ 0.08 | 26.07 $\pm$ 0.03 |

As you can see that our bound is again tighter than the Rademacher complexity based one under all settings.

(2) **Varying graph families, degrees, depth, etc.**

We consider 6 synthetic datasets that correspond to 6 graph families as shown below. All datasets have: # graphs = 200, # nodes per graph = 100, # classes = 2, train/test = 90%/10%, and random Gaussian node feature with unit 2 norm and 16 dimensions. All class labels of individual graphs are generated following the uniform distribution.

Erdos-Renyi-1 (ER-1): edge probability = 0.1, max-node-degree = 25
Erdos-Renyi-2 (ER-2): edge probability = 0.3, max-node-degree = 48
Erdos-Renyi-3 (ER-3): edge probability = 0.5, max-node-degree = 69
Erdos-Renyi-4 (ER-4): edge probability = 0.7, max-node-degree = 87
Stochastic-Block-Model-1 (SBM-1): two blocks, sizes = [40, 60], probs = [[0.25, 0.13], [0.13, 0.37]], max-node-degree = 25
Stochastic-Block-Model-2 (SBM-2): three blocks, sizes = [25, 25, 50], probs = [[0.25, 0.05, 0.02], [0.05, 0.35, 0.07], [0.02, 0.07, 0.40]], max-node-degree = 36

Since these generated datasets essentially require GNNs to fit to random labels which is arguably hard, we extend the number of training epochs to $200$ and use Adam as the optimizer with learning rate set to $1.0e^{-2}$.
More details can be found in the updated appendix. The log bound values are shown in the table below.

|  **l=2**   |`      ER-1      `|`      ER-2      `|`      ER-3      ` | `      ER-4      ` | `      SBM-1      ` | `      SBM-2      ` |
|------------|:----------------:|:------------------:|:---------------------:|:------------------:|:------------------:|:------------------:|
PAC-Bayes | **15.38** $\pm$ 0.12 | **15.13** $\pm$ 0.13 | **14.86** $\pm$ 0.25 | **14.69** $\pm$ 0.24 | **15.23** $\pm$ 0.12 | **15.35** $\pm$ 0.10 |
Rademacher| 17.37 $\pm$ 0.16 | 17.98 $\pm$ 0.13 | 18.15 $\pm$ 0.15 | 18.35 $\pm$ 0.10 | 17.88 $\pm$ 0.11 | 17.71 $\pm$ 0.09 |
**l=4**  |  |  |  |  |  |  |
PAC-Bayes | **27.00** $\pm$ 0.04 | **28.32** $\pm$ 0.07 | **29.18** $\pm$ 0.12 | **29.70** $\pm$ 0.14 | **28.14** $\pm$ 0.05 | **27.74** $\pm$ 0.04 |
Rademacher| 27.92 $\pm$ 0.02 | 29.57 $\pm$ 0.12 | 30.64 $\pm$ 0.18 | 31.34 $\pm$ 0.20 | 29.35 $\pm$ 0.14 | 28.87 $\pm$ 0.07 |
**l=6**  |  |  |  |  |  |  |
PAC-Bayes | **36.85** $\pm$ 0.25 | **39.65** $\pm$ 0.14 | **41.30** $\pm$ 0.22 | **42.24** $\pm$ 0.34 | **39.50** $\pm$ 0.17 | **38.63** $\pm$ 0.17 |
Rademacher| 37.10 $\pm$ 0.29 | 40.22 $\pm$ 0.19 | 42.00 $\pm$ 0.26 | 43.08 $\pm$ 0.39 | 40.04 $\pm$ 0.25 | 39.02 $\pm$ 0.19 |
**l=8**  |  |  |  |  |  |  |
PAC-Bayes | 46.79 $\pm$ 0.48 | **51.02** $\pm$ 0.21 | **53.10** $\pm$ 0.36 | **54.67** $\pm$ 0.38 | **50.44** $\pm$ 0.16 | **49.22** $\pm$ 0.36 |
Rademacher| **46.72** $\pm$ 0.51 | 51.16 $\pm$ 0.21 | 53.44 $\pm$ 0.39 | 55.06 $\pm$ 0.38 | 50.60 $\pm$ 0.17 | 49.29 $\pm$ 0.34 |

From the table, we can see that our bound is tighter than the Rademacher one under all 6 graph generative models with varying degree statistics and network depth except for a single setting (ER-1 with l=8 where d is the smallest). This again verifies our finding that our bound has a better dependence on the max-node-degree and tends to be tighter than the Rademacher complexity based one if d is large. We include more discussions in the paper and appendix.

---

### Decision · Program_Chairs · 2021-01-07
**Final Decision**

**Decision:**

Accept (Poster)

**Comment:**

This paper gives a new PAC-Bayesian generalization error bound for graph neural networks (GCN and MPGNN). The bound improves the previously known Rademacher complexity based bound given by Garg et al. (2020). In particular, its dependency on the maximum node degree and the maximum hidden dimension is improved.

This paper gives an interesting improvement on the generalization analysis of GNNs. The writing is clear, where its connection to existing work and its technical contribution are well discussed.
The biggest concern is its technical novelty. Indeed, the proof follows the out-line of Neyshabur et al. (2017). Given that the technical novelty would be a bit limited, however, the analysis should properly deal with the complicated structure specific to GNNs which makes the analysis more difficult than usual CNN/MLP and requires subtle and careful manipulations.
In addition to that, the improvement of the generalization bound is valuable for the literature (while the improvement seems a bit minor for graphs with small maximum degree).

For these reasons, I recommend acceptance for this paper.